# Effect of Intraoperative Sedation with Dexmedetomidine Versus Propofol on Acute Postoperative Pain Following Major Foot Surgery under Popliteal Sciatic Nerve Block: A Randomized Controlled Trial

**DOI:** 10.3390/jcm9030654

**Published:** 2020-02-28

**Authors:** RyungA Kang, Ji Won Choi, Ki-Sun Sung, Wongook Wi, Tae Soo Hahm, Hyun Sung Cho, Mi Kyung Yang, Justin Sangwook Ko

**Affiliations:** 1Department of Anesthesiology and Pain medicine, Samsung Medical Center, Sungkyunkwan University School of Medicine, Seoul 06351, Korea; ryunga.kang@samsung.com (R.K.); jiwon0715.choi@samsung.com (J.W.C.); wongook.wi@samsung.com (W.W.); ts.hahm@samsung.com (T.S.H.); chohs.cho@samsung.com (H.S.C.); anes.yang@samsung.com (M.K.Y.); 2Department of Orthopedics, Samsung Medical Center, Sungkyunkwan University School of Medicine, Seoul 06351, Korea; kisun.sung@samsung.com

**Keywords:** anesthesia, dexmedetomidine, pain, postoperative, sciatic nerve block

## Abstract

Intravenous (IV) dexmedetomidine is reported to prolong analgesia following peripheral nerve blocks. Popliteal sciatic nerve block provides effective postoperative analgesia, but some patients still experience severe pain during the early postoperative period. We aimed to evaluate the postoperative analgesic effects of IV dexmedetomidine versus propofol in patients undergoing foot surgeries under popliteal sciatic nerve block. Forty patients were enrolled and randomly assigned to receive either IV propofol (*n* = 20) or IV dexmedetomidine (*n* = 20) for intraoperative sedation. All the patients received continuous popliteal sciatic nerve block. The corresponding drug infusion rate was adjusted to achieve a modified observer’s assessment of alertness/sedation score of 3 or 4. The primary outcome was postoperative cumulative opioid consumption during the first 24 h after surgery. Thirty-nine patients were analyzed. The median (interquartile ranges) postoperative cumulative opioid consumption during the first 24 h after surgery was significantly lower in the dexmedetomidine group (15 (7.5–16.9) mg) than in the propofol group (17.5 (15–25) mg) (*p* = 0.019). The time to first rescue analgesic request was significantly greater in the dexmedetomidine group than in the propofol group (11.8 ± 2.2 h vs. 10.0 ± 2.7 h, *p* = 0.030) without the prolonged motor blockade (*p* = 0.321). Intraoperative sedation with dexmedetomidine reduced postoperative opioid consumption and prolonged analgesic duration after a popliteal sciatic nerve block.

## 1. Introduction

Major foot surgery is associated with moderate to severe postoperative pain [1]. Recently, continuous popliteal sciatic nerve block with perineural local anesthetic infusion has emerged as an efficient component of acute pain management after various foot surgeries. Although continuous popliteal sciatic nerve block has many benefits, including effective postoperative analgesia, reduced opioid consumption, and improved patient satisfaction [2,3,4,5], a significant number of patients still experience severe pain during the first 24 h after surgery [4].

Dexmedetomidine is a potent α2-adrenoreceptor agonist with sedative and analgesic properties [6]. Intravenous (IV) dexmedetomidine as a sedative for surgery under regional anesthesia has been studied widely because it produces the unconsciousness similar to that of natural sleep without respiratory depression [7], and reduces postoperative pain and opioid consumption [8,9]. More recent studies assessed the effect of dexmedetomidine as an intraoperative sedative drug on the analgesic duration under spinal anesthesia [8,9], but the evidence of its efficacy on popliteal sciatic nerve block is still lacking.

Therefore, we aimed to compare the postoperative analgesic effects of intraoperative infusion of dexmedetomidine vs. propofol to provide sedation to patients undergoing major foot surgery under popliteal sciatic nerve block. We hypothesized that intraoperative dexmedetomidine sedation would decrease postoperative cumulative opioid consumption during the first 24 h after surgery.

## 2. Materials and Methods

All the subjects gave their informed consent for inclusion before they participated in the study. The study was conducted in accordance with the Declaration of Helsinki, and the protocol was approved by the Ethics Committee of the Samsung Medical Center (SMC 2018-07-145-001), and this trial was prospectively registered on the Clinical Trial Registry of Korea (KCT0003303, principal investigator: Justin Sangwook Ko) on 30 October 2018. We enrolled 40 adult patients. Inclusion criteria were: age 19 years and older, American Society of Anesthesiologists (ASA) Physical Status classification I to III, inpatients scheduled for elective unilateral major foot bone surgeries (hallux valgus osteotomy, 1st toe bunion correction operation, metatarsal osteotomies, and great toe arthrodesis) under popliteal sciatic nerve block between November 2018 and December 2019 at the Samsung Medical Center, Seoul, Korea. All the operations were performed by a single surgeon (KSS). Eligible patients were identified from the surgeon’s operating list and contacted the day prior to their surgery to inform them of the study protocol. We excluded patients who refused to participate in the study, and those with a history of cardiac, renal, or hepatic disease, preexisting neurological deficits in the lower extremities, contraindications to peripheral nerve block, or allergy to local anesthetics. Further, we excluded patients undergoing soft tissue procedures.

A staff member of the Samsung Medical Center who was not otherwise involved in the study performed computer-generated block randomization in a 1:1 ratio to the propofol group (*n* = 20) and the dexmedetomidine group (*n* = 20). Patient allocation to each group was concealed in an opaque envelope, which was only opened by one of the authors not involved in either anesthetic management or outcome assessment. Propofol and dexmedetomidine differ in color and injection method; thus, the anesthesiologist involved in the foot surgery was not blinded to the group assignment. All other research personnel, outcome assessors, and caregivers were blinded to group allocation.

Patients received no medication prior to arrival to a dedicated block room, where the popliteal sciatic nerve block was performed approximately one hour before the scheduled operation. After applying standard ASA monitoring and supplemental oxygen, IV midazolam (1–1.5 mg) was administered for anxiolysis. Ultrasound-guided popliteal sciatic nerve block was performed by experienced staff anesthesiologists (J.S.K. and R.A.K.) with the patient in the prone position. After identifying the bifurcation of the sciatic nerve into the tibial and common peroneal nerves using a 6–15 MHz high-frequency linear array transducer (X-Porte; Sonosite, Bothell, WA, USA), a 18G block needle (E-Cath PLUS; PAJUNK, Geisingen, Germany) was advanced with an in-plane approach in the lateral-to-medial direction. The final needle tip position was between the tibial and common peroneal nerves within the paraneural sheath (Figure 1).

Subsequently, 30 mL local anesthetic (15 mL 0.75% ropivacaine and 15 mL 2% lidocaine with 2.5 µg/mL epinephrine) was injected to expand the paraneural sheath before advancing the catheter 3–4 cm past the needle tip in a tibial component. The optimal position of the catheter tip within the paraneural sheath was confirmed by visualization of the perineural local anesthetic spread on injection through the catheter. The patients were then shifted to the supine position. Following block completion, sensory and motor blockades were assessed every 5 min for 30 min. Sensory block was evaluated over the dorsum of the foot and the middle of the sole of the foot (tibial nerve territory) using a cold alcohol swab, and the extent of sensory loss was graded on a three-point scale (2 = normal; 1 = diminished sensation; 0 = loss of sensation). For motor block, ankle plantar flexion and dorsiflexion were graded using a three-point scale (2 = normal; 1 = weakness; 0 = complete loss of power). Block success was defined as a sensory score of 0 within 30 min of local anesthetic injection. If incomplete block or block failure occurred, the patient was converted to general anesthesia.

After confirming successful block establishment, the patients were transferred to the operating room and the study drugs were administered. The dexmedetomidine group received dexmedetomidine at 1 µg·kg^−1^ of the loading dose over 10 min followed by 0.2–0.7 µg·kg^−1^·h^−1^ for maintenance. If the patients did not achieve an observer’s assessment of alertness/sedation scale (OAA/S) [8,10] score of 3–4 following the completion of dexmedetomidine loading or maintenance, a small dose of midazolam (0.5–1 mg) was administered intravenously to achieve OAA/S of 3–4. The propofol group received propofol at an effect site concentration of 0.5–2.0 µg·mL^−1^ via target-controlled infusion (Orchestra^®^; Fresenius Vial, Brezins, France). The propofol was also adjusted to maintain an OAA/S of 3–4. Dexmedetomidine administration was discontinued when subcutaneous and skin suturing was started, and propofol administration was discontinued at the end of surgery. Oxygen was supplied through the facial mask, and continuous end-tidal CO_2_ monitoring was performed during the sedation. Hypotension (a mean blood pressure decrease of more than 20% the pre-sedation value) was treated with 2.5–5 mg ephedrine, and bradycardia (a heart rate lower than 50 beats per minutes) was treated with 0.5 mg atropine. Respiratory depression was defined as an oxygen saturation lower than 90%.

After surgery, all patients were transferred to the post-anesthesia care unit, where they remained until they met the discharge criteria [11]. Patients were then transferred to the surgical ward for further care. Pain in the foot at rest was measured using an 11-point numerical rating scale (NRS, 0–10; 0 = no pain, 10 = worst pain). Postoperative supplemental analgesia was standardized as follows: at the first report of pain at the surgical site (pain score > 0/10), in either the post-anesthesia care unit or the surgical ward, patient-controlled regional analgesia (PCRA) with 0.15% ropivacaine was initiated by a nurse who was blinded to the group allocation. The PCRA pump was programmed to deliver a continuous infusion of 8 mL·hour-1 and a bolus dose of 4 mL on demand with a lockout interval of 30 min. Both the start time of the PCRA and cumulative ropivacaine dose were recorded automatically by the pump (Gemster, Hospira, Inc., Lake Forest, IL, USA), and these data were later downloaded for analysis. Once oral intake was tolerated, all patients received oral tapentadol 50 mg (Nucynta^®^ ER TAB 50 mg, Janssen Korea, Seoul, Korea) every 12 h. Patients with persistent pain scores > 4/10 despite this regimen received rescue analgesia with either 10 mg IV morphine or 50 mg IV pethidine. PCRA was discontinued on postoperative day 2. All the patients remained in the hospital for 4 days after surgery and were followed-up at the outpatient clinic on postoperative day 14.

The primary outcome was cumulative postoperative opioid consumption at 24 h postoperatively (reported as IV morphine equivalents [12]). Secondary outcomes included resting pain score at 24 h, time to first rescue analgesic request, time to first toe movement, PCRA consumption at 24 h, incidence of nausea, vomiting, pruritus within 24 h, patient satisfaction with analgesia at 24 h, and quality of sleep on the first night. Patient satisfaction at 24 h, and quality of sleep were measured using a Likert scale (1 = very dissatisfied, 2 = dissatisfied, 3 = neutral, 4 = satisfied, and 5 = very satisfied). Residual block-related neurological symptoms (persistent numbness, paresthesia, or weakness) were also measured at 24 h, 48 h, and 14 days postoperatively. All the assessments and data collection were performed by the research personnel blinded to group allocation.

Based on the preliminary analysis (unpublished), the mean (standard deviation) cumulative opioid consumption at 24 h after foot surgery under popliteal sciatic nerve block was 19 (4) mg. A 20% reduction in cumulative opioid consumption with the use of intraoperative dexmedetomidine was considered clinically significant. With α = 0.05 and power of 80%, 19 patients were required in each group. Assuming a 5% dropout rate, we decided to enroll 20 patients per group. After confirming the normality of data distribution using the Shapiro–Wilk test, descriptive statistics and outcome variables were compared using the *t*-test or the Mann–Whitney test for continuous variables, and the chi-squared test or the Fisher’s exact test for categorical variables. Continuous variables are presented as mean ± standard deviation, or median and interquartile range as appropriate. Categorical variables are presented as numbers and percentages. Data analysis was performed using SPSS software (ver. 25.0; SPSS Inc., Chicago, IL). For all analyses, two-sided tests were used, and *p* < 0.05 was considered significant.

## 3. Results

### 3.1. Study Participants

From November 2018 to December 2019, 44 patients scheduled for major foot surgery under popliteal sciatic nerve block were assessed for eligibility, and four patients who did not meet the inclusion criteria were excluded (Figure 2).

All enrolled patients (*n* = 40) were randomly assigned to one of the two treatment groups (*n* = 20 each). One patient allocated to the propofol group was excluded after receiving the popliteal sciatic nerve block, because the surgical plan was changed to bilateral feet, and the anesthetic plan was changed to general anesthesia. This patient did not participate further in the study. A total of 39 patients (the propofol group (*n* = 19) and the dexmedetomidine group (*n* = 20)) completed the study and were included in the final analysis. The baseline patient and surgical characteristics are shown in Table 1. All the blocks met the criteria for success at 30 min following block completion, and none of the patients required supplemental analgesia.

### 3.2. Perioperative Parameters

During intraoperative sedation, none in the propofol group and 16 patients (80%) in the dexmedetomidine group required additional midazolam (Table 2). Other perioperative outcomes, including the incidence of respiratory depression, intraoperative hypotension, and bradycardia, were not significantly different between the two groups (Table 2). In the post-anesthetic care unit (PACU), pain scores were 0 and OAA/S scores were 5 in both groups, and the duration of the PACU stay was similar between the two groups (Table 2). No other significant adverse events occurred in either group during intraoperative sedation.

### 3.3. Study Outcomes

The median (interquartile range) postoperative cumulative opioid consumption during the first 24 h after surgery was significantly lower in the dexmedetomidine group (15 (7.5–16.9) mg) than in the propofol group (17.5 (15–25) mg) (*p* = 0.019) (Table 3). Time to first rescue analgesic request was significantly greater in the dexmedetomidine group compared to the propofol group (11.8 ± 2.2 h vs. 10.0 ± 2.7 h, *p* = 0.030). Time to first toe movement, pain scores at 24 h, PCRA consumption at 24 h, and the incidence of nausea, vomiting, and pruritus within 24 h were similar between the two groups (Table 3). Subjects receiving dexmedetomidine reported significantly better quality of sleep on the first postoperative night (2 (1.3–3) vs. 1 (1–2), *p* = 0.019). None of the patients in both groups reported popliteal sciatic nerve block-related complications within 14 postoperative days.

## 4. Discussion

In this randomized controlled trial, intraoperative IV dexmedetomidine sedation resulted in significant postoperative opioid-sparing effects compared with propofol sedation in patients undergoing major foot surgery with continuous popliteal sciatic nerve block. Moreover, IV dexmedetomidine resulted in prolonged analgesia without prolonged motor blockade and better sleep quality on the first postoperative night. Dexmedetomidine-related side effects, such as hypotension and bradycardia and prolonged recovery time, were not observed.

Recently, intraoperative sedation with IV dexmedetomidine has been reported to demonstrate significant reduction in the postoperative opioid consumption and prolongation of postoperative analgesia after both spinal anesthesia [9] and peripheral nerve blocks [13,14]. The exact underlying mechanism of action of IV dexmedetomidine is still not fully understood, but more recent data implicated varied pathways for sedation and analgesia [15]. The sedative effect of dexmedetomidine is purported to be mediated by the ascending noradrenergic pathway in the locus coeruleus [16], and the analgesic effect is mediated by the binding of the drug to the α2-adrenergic receptor-dependent descending pathway in the spinal cord [17] and the locus coeruleus [16], thereby blocking pain signal transmission by inhibiting the release of nociceptive transmitters [16,18]. In addition, pharmacokinetics and pharmacodynamics studies have suggested that, compared with perineural injection, an adequate or a relatively high dose of IV dexmedetomidine is required to provide clinical analgesic effects [6,19,20]. Previous studies have shown that the clinical analgesic effects of IV dexmedetomidine were not evident below certain cut-off values [14]; in particular, a comparative study on analgesic effects showed that dexmedetomidine showed less analgesic effect up to plasma concentrations of 2.4 ng/ml than remifentanil, and this can be extrapolated to the IV dexmedetomidine dose of 1.25 µg/kg [19,20]. In agreement therewith, in the present study, IV dexmedetomidine was administered at the average dose of 1.66 µg/kg during surgery, which is higher than the cut-off values suggested in previous studies; in part, this might be due to the significantly prolonged analgesia as marked by a longer time to first rescue analgesic request and a subsequent decrease in the postoperative opioid consumption in the dexmedetomidine group in comparison with the propofol group. Furthermore, compared with perineural injections, IV dexmedetomidine has been reported to selectively prolong duration of the sensory block without the prolonged motor block [13,14]. Thus, in our study, the IV dexmedetomidine group did not show increased motor block duration compared with the propofol group. This may be regarded as an added benefit for earlier ambulation of patients receiving regional anesthesia and analgesia.

Propofol acts on the gamma-aminobutyric acid and is widely used as an intraoperative sedative agent, because it is highly effective, allows rapid induction and emergence, and can be easily titrated to the required sedation level [21], whereas dexmedetomidine-based sedation is induced through the activation of α2-receptors in the locus coeruleus leading to a unique sedative state resembling natural sleep. Therefore, patients remain easily rousable, and respiration is minimally affected, with a significantly reduced risk of airway obstruction and respiratory depression [22]. In line with these findings, in our study, the incidence of respiratory depression was not significantly different between the two groups. However, a longer time is often required to achieve targeted levels of sedation with dexmedetomidine compared with propofol [23]. Similarly, in our study, a significantly higher number of patients in the dexmedetomidine group received additional midazolam to expedite the onset of sedation. However, the effect of midazolam on analgesic outcomes might be considered very minor, because it has sedative properties only; further, only a small dose of midazolam was used (1.1 ± 0.7 mg) in the dexmedetomidine group.

The most frequent side effects of dexmedetomidine are hemodynamic alterations, including hypertension, bradycardia, and hypotension, resulting from presynaptic and postsynaptic α2-receptor activations leading to vasoconstriction, baroreflex-mediated parasympathetic activation, and sympatholysis, respectively [24]. In our study, there were no significant differences in the incidences of intraoperative hypotension and bradycardia between the two groups. Prolonged sedation is another potential adverse effect of dexmedetomidine administration and is reported to show dose-dependent sedative effects [19]. In our study, however, the duration of sedation as reflected by the length of stays in the recovery room was not prolonged in the dexmedetomidine group compared with the propofol group. This may be explained by the fact that sedation usually resolves within 2 h after the infusion [25], which corresponds to the average time from initiation of the drug to discharge from the PACU. Overall, these aspects of dexmedetomidine should make it an attractive sedative drug for use in various regional anesthesia procedures, especially peripheral nerve blocks.

The main limitation of our study was that we did not assess pain scores and cumulative opioid consumptions within 24 h postoperatively. However, the time to first rescue analgesic request was assessed between the two groups, and this may be a more practical parameter to determine the analgesic effect of dexmedetomidine than the pain score assessment at set postoperative intervals. Another limitation is that the sedation level was measured using only the OAA/S score. Although the OAA/S score is a well-established method to evaluate sedation, it has the disadvantage of frequent patient stimulation, which may alter the actual level of sedation. Therefore, it would have been helpful to use a combination of the bispectral index (BIS) and the OAA/S score. However, care should be taken when using BIS values during dexmedetomidine sedation, because BIS values were reportedly lower with dexmedetomidine sedation than with propofol at comparable OAA/S scores [26].

## 5. Conclusions

In conclusion, our results demonstrated that intraoperative dexmedetomidine sedation significantly reduced postoperative opioid use over the first 24 h after operation and prolonged sensory block duration without prolonging motor block duration in the patients undergoing major foot surgery under popliteal sciatic nerve block. Based on these findings, IV dexmedetomidine may be considered a valuable sedative agent to be used in surgeries under popliteal sciatic nerve block.

## Figures and Tables

**Figure 1 jcm-09-00654-f001:**
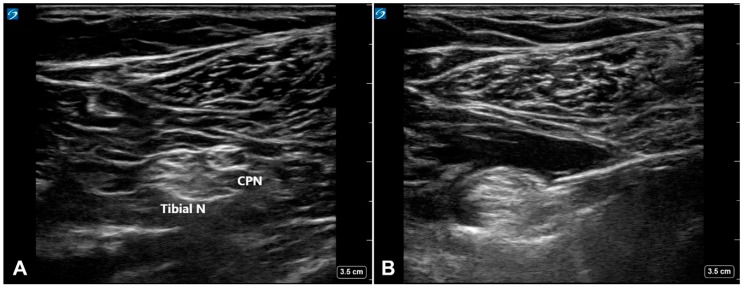
(**A**) Transverse ultrasound view of the sciatic nerve bifurcation in the popliteal fossa. (**B**) The needle tip located within the paraneural sheath.

**Figure 2 jcm-09-00654-f002:**
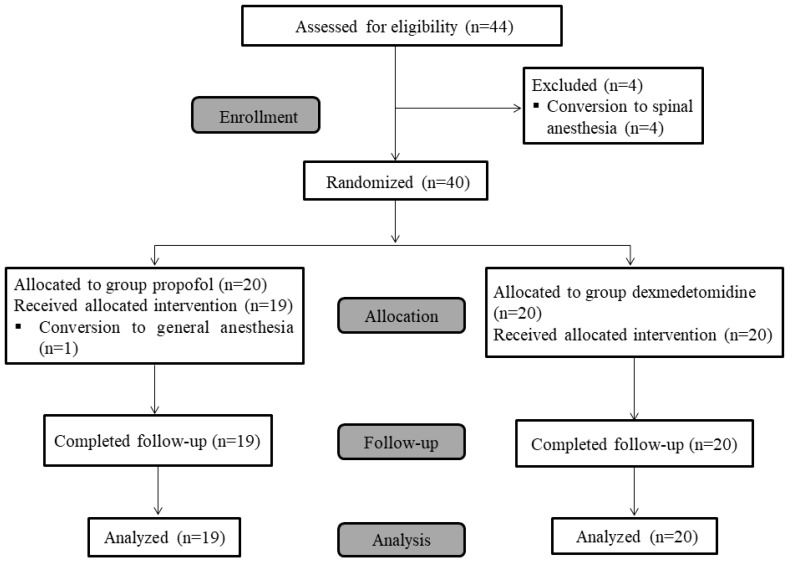
Consolidated Standards of Reporting Trials flow diagram showing patient progress through the study phases.

**Table 1 jcm-09-00654-t001:** Patient characteristics in the propofol group and in the dexmedetomidine group.

Parameter	Propofol (*n* = 19)	Dexmedetomidine (*n* = 20)	** p* Value
Age (years)	53.6 ± 12.7	49.8 ± 15.8	0.410
Sex (male/female)	3/16	4/16	>0.99
Height (cm)	160.1 ± 6.8	160.6 ± 9.2	0.846
Weight (kg)	60.1 ± 10.9	62.4 ± 12.4	0.540
ASA physical status (I/II)	8/11	13/7	0.205
Type of surgery			0.532
Hallux valgus osteotomy	16	16	
1^st^ toe bunion correction operation	2	1	
1^st^ metacarpophalangeal joint fusion	1	3	

Values are mean ± standard deviation or number. The *p* value for the *t*-test or the chi-squared test is set at 0.05. ASA, American Society of Anesthesiologists.

**Table 2 jcm-09-00654-t002:** Perioperative parameters in the propofol group and in the dexmedetomidine group.

Parameters	Propofol (*n* = 19)	Dexmedetomidine (*n* = 20)	** p* Value
**In operating room**			
Total infused amount of study drug	211.6 ± 77.1 mg	103.7 ± 37.6 µg	N/A
Requirement for additional midazolam, *n*	0	16 (80%)	<0.001
Dose of midazolam, mg	0	1.1 ± 0.7	<0.001
Respiratory depression, *n*	0	0	>0.99
Hypotension (requiring ephedrine treatment), *n*	1	1	>0.99
Bradycardia, *n*	1	1	>0.99
Mean intraoperative blood pressure, mm Hg	77.9 ± 11.0	78.0 ± 7.9	0.978
Mean intraoperative heart rate, beats per minute	75.7 ± 10.7	72.0 ± 11.5	0.309
Duration of surgery (minutes)	59.6 ± 19.5	58.8 ± 23.3	0.899
**In PACU**			
Static pain score at PACU, (0–10)	0 (0–0)	0 (0–0)	0.607
OAA/S score	5 (5–5)	5 (5–5)	0.411
Nausea/vomiting, *n*	0	0	N/A
Pruritus, *n*	0	0	N/A
Duration of PACU stay (minutes)	48.6 ± 11.6	46.9 ± 7.6	0.595

Values are mean ± standard deviation, median (interquartile ranges), or number (percentage). *The *p* value for the *t*-test, the Mann–Whitney U-test, or the chi-squared test is set at 0.05. N/A, not applicable. OAA/S, observer’s assessment of alertness/sedation. PACU, post-anesthetic care unit.

**Table 3 jcm-09-00654-t003:** Clinical outcome differences between the propofol group and the dexmedetomidine group.

Outcomes	Propofol (*n* = 19)	Dexmedetomidine (*n* = 20)	** p* Value
**Primary outcome**			
Cumulative opioid consumption at 24 h: converted to IV morphine equivalent, mg	17.5 (15–25)	15 (7.5–16.9)	0.019
**Secondary outcomes**			
Time to first rescue analgesic request, h	10.0 ± 2.7	11.8 ± 2.2	0.030
Time to first toe movement, h	11.2 ± 2.5	12.1 ± 2.9	0.321
Static pain score at 24 h, (0-10)	3.5 (2–5)	3 (2–4.8)	0.607
PCRA consumption at 24 h, mL	143.2 ± 64.5	115.8 ± 40.6	0.125
Nausea within 24 h, *n*	13 (68.4%)	7 (35%)	0.056
Mild/moderate/severe	10/0/3	6/0/1	0.065
Vomiting within 24 h, *n*	5 (26.3%)	1 (5%)	0.091
Pruritus within 24 h, *n*	0	0	N/A
Quality of sleep on the first night (^†^ Likert scale; 1 to 5)	1 (1–2)	2 (1.3–3)	0.019
Patient satisfaction with pain relief at 24 h (^†^ Likert scale; 1 to 5)	2 (1–3)	3 (2–3)	0.050

Values are mean ± standard deviation, median (interquartile range), or number (percentage). * The *p* value for the *t*-test, the Mann–Whitney U-test, or the chi-squared test is set at 0.05. ^†^ A Likert scale, where 1 = very dissatisfied, 2 = dissatisfied, 3 = neutral, 4 = satisfied, and 5 = very satisfied. PACU, post-anesthetic care unit; PCRA, patient-controlled regional analgesia; N/A, not applicable.

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
