# Peer review of "Effect of Intraoperative Sedation with Dexmedetomidine Versus Propofol on Acute Postoperative Pain Following Major Foot Surgery under Popliteal Sciatic Nerve Block: A Randomized Controlled Trial"

_jcm, 2020, doi:10.3390/jcm9030654_

Round 1

Reviewer 1 Report

Dear author,

Thank you very much for your effort to produce a valuable article.

However there are some issues that need to be clarified in order to better understand this paper.

The title of this article is "Effect of Intraoperative Sedation with Dexmedetomidine Versus Propofol on Acute Postoperative Pain Following Major Foot Surgery under Popliteal Sciatic Nerve Block". The authors conclude that intraoperative dexmedetomidine sedation provides postoperative opioid-sparing effects during the first 24 hours after surgery in patients undergoing major foot surgery under popliteal sciatic nerve block. Moreover, dexmedetomidine sedation was superior as it prolonged sensory block duration without prolonging motor block duration and preserved respiratory function as well".

1-This article most likely would examine post operative pain. One of the main outcomes of this study is also to determine the cumulative postoperative analgesic consumption at 24 hours postoperatively. Understandably, both cumulative ropivacaine dose (boluses and maintenance) and opioids as rescue analgesics were recorded and compared in two groups receiving dexmedetomidine and propofol. The authors admit that the pain scores were not assessed. It is difficult to understand why pain scores were not measured in a study that would like to investigate post operative pain and on which base analgesics were encouraged to be administered. Would the authors kindly clarify these issues?

2-The study protocol suggest for authors to adjust the loading dose and maintenance infusion of dexmedetomidine and propofol in order to achieve an observer’s assessment of alertness/sedation scale score of 3–4. However table 2 demonstrates median and interquartile range score of 5. Why this was difficult to achieve at least in the propofol group? Can authors explain this?

3- Please explain how did the authors conclude the superiority of patients in the dexmedetomidine group in terms of respiratory preservation as no one in either group did experience sedation-related respiratory complications.

4- The authors state that "in the present study, IV dexmedetomidine was administered at an average dose of 1.66 mcg/kg during surgery, which is higher than the cut-off values suggested in previous studies; this might be due in part to the significantly prolonged analgesia, as marked by a longer time to first rescue analgesic request". Can the author please confirm or explain that generally a higher dose of dexmedetomidine is required in order to achieve an appropriate sedation level than the dose required for analgesic purposes?

5- In table 1 the abbreviation for PACU may be deleted as PACU is not used in this table.

6- In table 2. Can authors explain why they did not statistical analyse on requirement for additional midazolam administration and also midazolam dose? To my opinion this would be sensible to do.

7- In spite of having midazolam in addition to the relatively high dose of dexmedetomidine on board, I'm very surprised that the post operative length of stay in the PACU was not prolonged in the dexmedetomidine group. Can the authors explain this controversy with the common practice?

Look forward to receiving the revision version of this article

Author Response

Point 1: This article most likely would examine postoperative pain. One of the main outcomes of this study is also to determine the cumulative postoperative analgesic consumption at 24 hours postoperatively. Understandably, both cumulative ropivacaine dose (boluses and maintenance) and opioids as rescue analgesics were recorded and compared in two groups receiving dexmedetomidine and propofol. The authors admit that the pain scores were not assessed. It is difficult to understand why pain scores were not measured in a study that would like to investigate post-operative pain and on which base analgesics were encouraged to be administered. Would the authors kindly clarify these issues?

Response 1: Thank you for your valuable comments. In addition to cumulative ropivacaine dose and opioids, we assessed the pain scores at postoperative 24 h (secondary outcome) and they were similar between the dexmedetomidine and propofol groups (3 [2-4.8] vs. 3.5 [2-5], P = 0.607, Table 3). In the limitation, we admitted that we did not assess the pain scores and cumulative opioid consumptions within 24 hours postoperatively. While designing this study, we assigned the cumulative opioid consumption at postoperative 24 h as our primary outcome instead of the pain score because we deemed the former as more objective analgesic evaluation of IV dexmedetomidine.

Point 2: The study protocol suggests for authors to adjust the loading dose and maintenance infusion of dexmedetomidine and propofol in order to achieve an observer’s assessment of alertness/sedation scale score of 3–4. However, table 2 demonstrates median and interquartile range score of 5. Why this was difficult to achieve at least in the propofol group? Can authors explain this?

Response 2: Thank you for your valuable comments. The OAA/S scores in Table 2 was measured in the PACU, not in the operating room, as indicated under In PACU heading. During the operation, the patients in both groups were managed to achieve an OAA/S score of 3-4.

Point 3: Please explain how did the authors conclude the superiority of patients in the dexmedetomidine group in terms of respiratory preservation as no one in either group did experience sedation-related respiratory complications.

Response 3: We agree with your opinion and have deleted “and preserved respiratory function as well” in the conclusion.

Point 4: The authors state that "in the present study, IV dexmedetomidine was administered at an average dose of 1.66 mcg/kg during surgery, which is higher than the cut-off values suggested in previous studies; this might be due in part to the significantly prolonged analgesia, as marked by a longer time to first rescue analgesic request". Can the author please confirm or explain that generally a higher dose of dexmedetomidine is required in order to achieve an appropriate sedation level than the dose required for analgesic purposes?

Response 4: Thank you for your valuable comments. As mentioned in the Discussion section, previous studies have shown that the clinical analgesic effect of IV dexmedetomidine was not dose-dependent and was not evident below certain cut-off values, and a comparative study on analgesic effects showed that 2.4 ng/ml of dexmedetomidine (extrapolated to 1.25 mcg/kg) produced less analgesic effect compared to remifentanil. In our study, the average dose of dexmedetomidine administered for the sedation was 1.66 mcg/kg, which is higher than 1.25 mcg/kg, and this might be the reason for the observed prolonged analgesia in the dexmedetomidine group. Our results mainly support the prolonged analgesic effect rather than the dose required for the sedative purposes, and thus, based on our findings, it is difficult to explain or confirm that higher dose of dexmedetomidine is required to achieve an appropriated sedation level than the dose required for the analgesic purposes.     

Point 5: In table 1 the abbreviation for PACU may be deleted as PACU is not used in this table.

Response 5: Thank you for your comment. We have deleted the abbreviation for PACU in Table 1.

Point 6: In table 2. Can authors explain why they did not statistical analyse on requirement for additional midazolam administration and also midazolam dose? To my opinion this would be sensible to do.

Response 6: Thank you for your comments. We added the results of the P values in Table 2.

Point 7: In spite of having midazolam in addition to the relatively high dose of dexmedetomidine on board, I'm very surprised that the postoperative length of stay in the PACU was not prolonged in the dexmedetomidine group. Can the authors explain this controversy with the common practice?

Response 7: Thank you for your valuable comments. One of major concerns after dexmedetomidine infusion is the prolonged recovery time and the resultant delay in PACU discharge time. To prevent the possible delayed recovery time, we discontinued dexmedetomidine administration when subcutaneous and skin suturing was started, which was about 15-20 minutes earlier than when we discontinued propofol. As a result, we did not observe a delayed recovery time in the dexmedetomidine group as evidenced by similar durations of PACU stay between two groups. We added following sentence in the Methods; Dexmedetomidine was discontinued when subcutaneous and skin suturing was started, and propofol administration was discontinued at the end of surgery.”

----------------------------The end-------------------------
We hope the revised manuscript will better meet the requirements of your journal for publication. We thank the Editors and the Reviewers of Journal of clinical medicine once again for the constructive review of our paper.

With regards,

Dr. Justin Sangwook Ko

Justin Sangwook Ko, M.D., Ph. D., Department of Anesthesiology and Pain Medicine, Samsung Medical Center, 81 Irwon ro, Gangnam gu, Seoul 06351, Korea. Tel.: +82-2-3410-2454; Fax: +82-2-3410-0361; E-mail: [email protected]

Reviewer 2 Report

The subject of the article is important. However the clinical Surgery intervention must be better explain. Why the Foot Surgery under Popliteal Sciatic Nerve Block?

Moreover, some minor points must be improved or corrected:

The Abstract is good, however need a sentence about the clinical importance of the work.

Why the author excluded patients undergoing soft tissue procedures? please insert inclusion criteria in the Materials and Methods section

In my opinion the preliminary analysis (unpublished) should go to the results section.

The results need to be better explained the Perioperative parameters, and Outcomes and the scales apply?

What are the ASA physical status, Chevron osteotomy and Lapidus operation? Pleased explain in the paper

The conclusions need to be better explained.

Author Response

Response to Reviewer 2 Comments

Point 1: The subject of the article is important. However the clinical Surgery intervention must be better explain. Why the Foot Surgery under Popliteal Sciatic Nerve Block?

Response 1: Thank you for your comments. In our hospital, we perform foot surgery under popliteal sciatic nerve block for both surgical anesthesia and postoperative analgesia. Actually, other previously published study performed foot surgery under spinal anesthesia because of the use of thigh tourniquet.(Zaric et al. 2004) However, in our institution, we routinely use ankle tourniquet during hallux valgus osteotomy, 1st toe bunion correction operation, etc. Also, the popliteal sciatic nerve block is performed in a dedicated block room approximately one hour before the operation to achieve effective surgical anesthesia. We added “..popliteal sciatic nerve block was performed approximately one hour before the scheduled operation” in the Methods.

[Reference] Zaric, D., K. Boysen, J. Christiansen, U. Haastrup, H. Kofoed, and N. Rawal. 2004. 'Continuous popliteal sciatic nerve block for outpatient foot surgery--a randomized, controlled trial', Acta Anaesthesiol Scand, 48: 337-41.

Point 2: Moreover, some minor points must be improved or corrected:

The Abstract is good, however need a sentence about the clinical importance of the work.

Response 2: Thank you for your valuable comment. We have inserted a sentence about the clinical importance of the work as follows: Popliteal sciatic nerve block provides effective postoperative analgesia, but some patients still experience severe pain during early postoperative period.”

Point 3: Why the author excluded patients undergoing soft tissue procedures? please insert inclusion criteria in the Materials and Methods section

Response 3: Thank you for your comments. The patients undergoing soft tissue procedures were not included in the study because the severity of postoperative pain is less severe compared to the patients undergoing bony surgeries such as hallux valgus osteotomy and 1st toe bunion correction operations. As the reviewer suggested, we added the inclusion criteria in the Materials and Methods section as follows:

Inclusion criteria were: age 19 years and older, American Society of Anesthesiologists (ASA) Physical Status classification I to III, inpatients scheduled for elective unilateral major foot bone surgeries (hallux valgus osteotomy, 1st toe bunion correction operation, metatarsal osteotomies and great toe arthrodesis) under popliteal sciatic nerve block between November 2018 and December 2019 at Samsung Medical Center, Seoul, Korea.”

Point 4: In my opinion the preliminary analysis (unpublished) should go to the results section.

Response 4: Thank you for your comment. However, the preliminary analysis was based on the pilot study to calculate the sample size in present study. Therefore, we believe it is better to mention it in the methods section in the statistical analyses section.

Point 5: The results need to be better explained the Perioperative parameters, and Outcomes and the scales apply?

Response 5: Thank you for your suggestions. We described important perioperative parameters and outcomes in the results.

Point 6: What are the ASA physical status, Chevron osteotomy and Lapidus operation? Pleased explain in the paper.

Response 6: Thank you for your comments. ASA physical status classification system is a system for assessing the fitness of patients before surgery. These are: ASA physical status I (healthy person), II (mild systemic disease), III (severe systemic disease), IV (severe systemic disease that is a constant threat to life), V (a moribund person who is not expected to survive without the operation), and VI (a declared brain-dead person whose organs are being removed for donor purposes. To enhance the reader’s readability, Chevron osteotomy and Lapidus operation were changed to hallux valgus osteotomy and 1st toe bunion correction operation. We revised the Table 1.

Point 7: The conclusions need to be better explained.

Response 7: We agree with your opinion and revised the conclusion as follows:

“In conclusion, our results demonstrated that intraoperative dexmedetomidine sedation significantly reduced postoperative opioid use over the first 24 h after operation and prolonged sensory block duration without prolong motor block duration in patients undergoing major foot surgery under popliteal sciatic nerve block. Based on these findings, IV dexmedetomidine may be considered a viable sedative agent to be used in surgeries under popliteal sciatic nerve block.”

------------------------The end-------------------------
We hope the revised manuscript will better meet the requirements of your journal for publication. We thank the Editors and the Reviewers of Journal of Clinical Medicine once again for the constructive review of our paper.

With regards,

Dr. Justin Sangwook Ko

Justin Sangwook Ko, M.D., Ph. D., Department of Anesthesiology and Pain Medicine, Samsung Medical Center, 81 Irwon ro, Gangnam gu, Seoul 06351, Korea. Tel.: +82-2-3410-2454; Fax: +82-2-3410-0361; E-mail: [email protected]

Round 2

Reviewer 1 Report

Dear author,

Thank you for added comments and revision of this work. It is now much more understandable to the readers.

Please just pay attention to two minor comments.

In table 2 please provide the abbreviation for PACU.

Conclusion may need some clarification: please see in bold.

“In conclusion, our results demonstrated that intraoperative dexmedetomidine sedation significantly reduced postoperative opioid use over the first 24 h after operation and prolonged sensory block duration without prolong (or prolonging) motor block duration in patients undergoing major foot surgery under popliteal sciatic nerve block. Based on these findings, IV dexmedetomidine may be considered a viable (or valuable) sedative agent to be used in surgeries under popliteal sciatic nerve block.”

Congratulation with this paper

Author Response

Point 1. In table 2 please provide the abbreviation for PACU.

Response 1. Thank you for your comments. We have provided the abbreviation for PACU in Table 2.

Point 2. Conclusion may need some clarification: please see in bold.

“In conclusion, our results demonstrated that intraoperative dexmedetomidine sedation significantly reduced postoperative opioid use over the first 24 h after operation and prolonged sensory block duration without prolong (or prolonging) motor block duration in patients undergoing major foot surgery under popliteal sciatic nerve block. Based on these findings, IV dexmedetomidine may be considered a viable (or valuable) sedative agent to be used in surgeries under popliteal sciatic nerve block.”

Response 2. Thank you for your comments. We have revised the conclusion according to the reviewer’s suggestion.

-------------------The end---------------------------------------------------------

We thank the Editors and the Reviewers of Journal of clinical medicine once again for the constructive review of our paper.

With regards,

Dr. Justin Sangwook Ko

Justin Sangwook Ko, M.D., Ph. D., Department of Anesthesiology and Pain Medicine, Samsung Medical Center, 81 Irwon ro, Gangnam gu, Seoul 06351, Korea. Tel.: +82-2-3410-2454; Fax: +82-2-3410-0361; E-mail: [email protected]

Reviewer 2 Report

This article was revised appropriately.

I recommend accept            

Author Response

We thank the Editors and the Reviewers of Journal of clinical medicine once again for the constructive review of our paper.

With regards,

Dr. Justin Sangwook Ko

Justin Sangwook Ko, M.D., Ph. D., Department of Anesthesiology and Pain Medicine, Samsung Medical Center, 81 Irwon ro, Gangnam gu, Seoul 06351, Korea. Tel.: +82-2-3410-2454; Fax: +82-2-3410-0361; E-mail: [email protected]